# miR-1-5p targets TGF-βR1 and is suppressed in the hypertrophying hearts of rats with pulmonary arterial hypertension

**Martin Connolly** [1], **Benjamin E. Garfield**[1,2], **Alexi Crosby**[3], **Nick W. Morrell**[3], **Stephen J. Wort**[2], **Paul R. Kemp**[1]*

**1** Molecular Medicine, National Heart & Lung Institute, Imperial College London, London, United Kingdom, **2** National Pulmonary Hypertension Centre at the Royal Brompton and Harefield NHS Trust and National Heart and Lung Institute, Imperial College London, London, United Kingdom, **3** Department of Medicine, Addenbrooke's Hospital, University of Cambridge, Cambridge, United Kingdom

* p.kemp@imperial.ac.uk

**Data Availability Statement:** All relevant data are within the manuscript, full blots are provided as supplementary material.

## Abstract

The microRNA miR-1 is an important regulator of muscle phenotype including cardiac muscle. Down-regulation of miR-1 has been shown to occur in left ventricular hypertrophy but its contribution to right ventricular hypertrophy in pulmonary arterial hypertension are not known. Previous studies have suggested that miR-1 may suppress transforming growth factor-beta (TGF-β) signalling, an important pro-hypertrophic pathway but only indirect mechanisms of regulation have been identified. We identified the TGF-β type 1 receptor (TGF-βR1) as a putative miR-1 target. We therefore hypothesized that miR-1 and TGF-βR1 expression would be inversely correlated in hypertrophying right ventricle of rats with pulmonary arterial hypertension and that miR-1 would inhibit TGF-β signalling by targeting TGF-βR1 expression. Quantification of miR-1 and TGF-βR1 in rats treated with monocrotaline to induce pulmonary arterial hypertension showed appropriate changes in miR-1 and TGF-βR1 expression in the hypertrophying right ventricle. A miR-1-mimic reduced enhanced green fluorescent protein expression from a reporter vector containing the TGF-βR1 3'-untranslated region and knocked down endogenous TGF-βR1. Lastly, miR-1 reduced TGF-β activation of a (mothers against decapentaplegic homolog) SMAD2/3-dependent reporter. Taken together, these data suggest that miR-1 targets TGF-βR1 and reduces TGF-β signalling, so a reduction in miR-1 expression may increase TGF-β signalling and contribute to cardiac hypertrophy.

## Introduction

Pulmonary arterial hypertension (PAH) is characterised by high, pre-capillary pulmonary vascular resistance caused by remodelling of pulmonary arterioles [1,2]. PAH can be, heritable, idiopathic, related to drug or toxin exposure or associated with an underlying condition [3].

Across all aetiologies of PAH, the pathology remains largely consistent and involves significant restriction and remodelling of the pulmonary vasculature, leading to a reduction in cross-

**Funding:** Funded by Award recipient: Paul Kemp. Grant number: FS/14/71/31038 Funder: British Heart Foundation (BHF). Funder website: https://www.bhf.org.uk/. The funders had no role in study design, data collection and analysis, decision to publish, or preparation of the manuscript.

**Competing interests:** The authors have declared that no competing interests exist.

sectional area available for pulmonary blood flow [2]. The elevated vascular resistance promotes right ventricular (RV) hypertrophy [1] as a consequence of increased afterload and can lead to right heart failure and premature death [2].

As adult myocardial cells do not divide, increased RV mass is a consequence of cell hypertrophy rather than proliferation. Increased activity of a number of signalling pathways has been demonstrated to drive cardiomyocyte hypertrophy. One pathway shown to be increased in a range of different hypertrophic conditions is the transforming growth factor-beta (TGF-β) signalling pathway. For example, TGF-β expression is elevated in models of myocardial infarction and remodelling, where it is known to associate with fibrosis and hypertrophy [4]. To stimulate the pathway, TGF-β binds to a complex of type I and type II receptors increasing the serine/threonine kinase activity of the receptors with the consequent activation of the (mothers against decapentaplegic homolog) SMAD signalling pathway and activation of TGF-β activated kinase 1 (TAK1) [4]. Activation of the TGF-β signalling pathway is important in fibrosis and the synthesis of matrix components in the heart in response to myocardial infarction as well as to the inflammatory response of the heart [5,6]. TGF-β also contributes to cardiac myocyte hypertrophy in particular in response to increased angiotensin II [7]. Factors that modulate TGF- β signalling are therefore likely to contribute to hypertrophy.

MicroRNAs (miRNAs) are key regulators of cell phenotype that fine-tune the proteome by inhibiting the translation or promoting the degradation of target mRNAs. A number of miRNAs have been shown to modulate the expression of components of the TGF-β signalling pathway, both activating and inhibiting the pathway. miRNA activators of the pathway target inhibitory components and include miR-424-5p which targets SMAD7 and miR-542-5p, which targets SMAD7 and SMURF1 (SMAD specific E3 ubiquitin protein ligase 1) [8,9]. Inhibitors of the pathway include miR-101a which targets TGF-β receptor type I (TGF-βR1), and miR-422a which targets SMAD4, the co-SMAD required for both bone morphogenic protein (BMP) and TGF-β signalling [10,11]. Basal expression of these miRNAs will contribute to setting the level of protein synthesis and thereby the capacity of the cell to signal under basal conditions. To contribute to changes in cell phenotype, expression of these miRNAs also needs to change in response to the disease process. Consistent with increased TGF-β signalling contributing to hypertrophy the expression of miR-424-5p is increased in some models of cardiac hypertrophy, whereas the expression of miR-101a is suppressed post-coronary occlusion [12,13].

miR-1, one of the most highly expressed miRNAs in the heart, contributes to myocyte phenotype. miR-1 expression is suppressed in hypertrophy, where TGF-β signalling is increased [14]. Similarly, miR-1 is suppressed in the skeletal muscle of patients with intensive care unit acquired weakness (ICUAW) who have increased nuclear pSMAD2/3 [15] suggesting reduced miR-1 contributes to the increase in TGF-β signalling. To date studies have shown an indirect effect of miR-1 on TGF-β signalling as the miRNA targets and suppresses histone deacetylase 4 (HDAC4) expression. This suppression of HDAC4 protein content increases follistatin expression, thereby reducing the amount of free TGF-β ligand and suppressing TGF-β signalling [16]. However, bioinformatic analysis predicts that miR-1 will target the TGF-β type I receptor ALK5 (activin A receptor type II-like kinase) suggesting a more direct effect on the pathway. We therefore determined the expression of miR-1 and ALK5 in the right ventricle of rats with pulmonary hypertension as a consequence of monocrotaline treatment. This model allows us to investigate factors contributing to myocyte hypertrophy. We hypothesized that RV hypertrophy would be associated with a reduction in miR-1 and increased TGF-βR1 protein levels. We also hypothesized that miR-1 would suppress ALK5 and TGF-β signalling in cells in culture.

## Methods

### In silico analysis of miR-1

Potential targets in the TGF-β signalling pathway for miR-1 were identified via the miRWalk 2.0 database, [17]. Putative targets were identified but screening the data base for targets identified by all 10 of the available algorithms. Putative targets were chosen as those identified by 5 or more algorithms. TGF-βR1/ALK-5 was predicted as a target by 6 algorithms including miTRWalk, miRanda and Targetscan. As TGF-β signalling has previously been shown to be important in regulating cardiac size, it was chosen as the basis of an enquiry into miR-1 regulation of cardiac hypertrophy.

### Monocrotaline rat model

Monocrotaline (MCT, Sigma; C2401) or phosphate buffered saline (PBS) was administered to 6–8 week-old rats via s.c. injection as previously described [18]. Eighteen male Sprague Dawley rats bought from Charles River Laboratories, Harlow, UK were included in the study and were randomly assigned to monocrotaline or PBS treatment. The study was reviewed and approved by the Cambridge University Animal Welfare and Ethical Review Body under the Home office license number 80/2460 19b/Section E No 7. Animals were housed and monitored according to local guidelines in the Centre for Biomedical Resources at Cambridge University. Animals were housed in groups (2–4 per cage) in temperature-controlled accommodation with free access to water and food. A day night cycle was maintained throughout the course of treatment. A number of quality improvement projects, aimed at increasing the animals' welfare, were conducted during the project, from which the rats benefitted. Animals were checked for health by staff at least once per day and were weighed every 2–3 days during treatment. Any animal that showed signs of distress of moderate severity or more, or who lost 20% of body weight was managed as per protocol and humanely killed. Animals who reached the end of the treatment course were anaesthetised with isoflurane, to render them unconscious, before right heart catheterisation was performed. They were subsequently humanely killed, without regaining consciousness. Consequently, animals did not survive to require post-operative care. Animals were euthanised by exsanguination, by transecting the femoral vessels, under terminal anaesthesia, as per the project license. The primary outcomes for this part of the study were the expression of miR-1 and of Alk5.

### RNA extraction & RT-qPCR

Ventricular RNA was extracted as previously described [19]. Cell RNA extractions were performed using the TaKaRa CellAmp Direct RNA kit (Clontech) as per manufacturer's protocol. miRNA expression was quantified by reverse transcription followed by EXPRESS SyBr PCR using the Agilent Genomics kit for polyA addition and first strand synthesis according to the manufacturer's instructions. mRNA was quantified by reverse transcription (RT) followed by quantitative polymerase chain reaction (qPCR) using SYBR green detection as previously described [20] and the primer sequences as shown in Table 1. The reverse primer for the PCR of miR-1 and U6 was the reverse primer provided with the polyadenylation and first strand synthesis kit.

### Cell culture

LHCN-M2 human skeletal myoblasts obtained from Vincent Mouly (Sorbonne Unitversité) in 2014 and were cultured in Skeletal muscle growth medium (PromoCell) supplemented with

**Table 1. Primer list for qPCR.**

| Gene | Forward sequence | Reverse sequence |
|---|---|---|
| miR-1-5p | CCGGTGGAATGTAAAGAAGTATGTAT | Agilent Universal reverse primer |
| U6 (housekeeper) | CTCGCTTCGGCAGCACA | Agilent Universal reverse primer |
| TGF-βR1 | GAACTCCCAACTACAGAAAAGCA | GCAGACTGGACCAGCAATGA |
| GAPDH (housekeeper) | GGTGGTCTCCTCTGACTTCAACA | GTTGCTGTAGCCAAATTCGTTGT |

20% FCS were transfected with miRNA mimic (miRVana) and Lipofectamine 2000 (Invitrogen) as previously described [21].

## Protein isolation from cells & western blotting

Cells were lysed in 1X cell lysis buffer (Cell Signalling Technologies) supplemented with protease inhibitor cocktail (Sigma) (1/100) and quantified as previously described [19]. Protein expression was analysed by western blotting after electrophoresis through a 10% SDS polyacrylamide gel as previously described [22]. Blots were probed for ALK5 protein using anti-TGF-βR1 (Santa Cruz, sc-9048) diluted 1:500 in PBS supplemented with 5% milk overnight at 4˚C, washed 3 times for 5 mins per wash in PBS Tween20 (0.05%) and detected with Horseradish peroxidase conjugated anti-rabbit IgG (1:3000 in PBS +5% milk) for 90 mins. Uncropped original blots are shown in S1 and S2 Figs.

## EGFP reporter cloning & assay

A section of the 3'UTR of TGF-βR1 containing the putative miR-1 binding site was amplified from human cDNA using primers: forward (5'–3'), GGAGATCTGGGTGTTTGATATTTCTTCAT reverse: (5'-3') GGGGGATCCGGACATTTTCTGTACATATCTTA and ligated into pGEM-T Easy (Promega). After sequencing to ensure the correct DNA had been amplified the insert was removed by BglII and BamHI digestion and ligated into a pCAGGS-enhanced green fluorescent protein (EGFP) vector down-stream of the coding region. Final plasmid sequences were confirmed by sequencing.

To quantify the effect of miR-1 on protein expression cells were transfected with the miR-mimic or control as previously described [21]. Twenty four hours later, the cells were transfected with this pCAGGS-EGFP-reporter either with or without the 3'UTR inset, cells were then lysed 24h later and EGFP expression was quantified by fluorescence with excitation at 480nm and emission detected at 510nm in a Cytofluor plate reader (Applied Biosystems).

## Luciferase assay

24 hours following miRNA transfection, cells were transfected with a 3:1 ratio of luciferase reporter vectors (CAGA)12 (firefly luciferase) and pRLTK (renilla luciferase) as previously described; the (CAGA)12 vector contains a SMAD binding element (SBE) specific to SMAD3 and SMAD4 [23]. 24 hours following this transfection cells were left untreated or treated with 5ng or 1ng of TGF-β1 ligand for 6 hours before harvesting for luciferase assay as previously described [9].

## Statistics

All statistical analyses were performed in GraphPad PRISM and no samples were excluded as outliers and data were analysed using a between samples design. Animal experiments were conducted in two groups for a total of 9 MCT- and 9 PBS-treated animals. Differences in

miRNA and mRNA expression between animal groups and left and right ventricles were calculated via Kruskal-Wallis test (ANOVA) for non-parametric data with post-hoc analysis using Dunn's multiple test correction. All other comparisons calculated using Student's t-test for normally distributed data or by Mann-Whitney U test for non-parametric data. *In vitro* mRNA expression and luciferase data shown were produced in three independent experiments, each consisting of six independent transfections; mRNA measures assayed in duplicate. Box-plots expressed as median with min-max bars. *In vitro* protein expression data shown via western blot are three independent experiments from 6-well plates. All tests were based on two tailed analysis and differences were taken to be significant at $p < 0.05$.

## Results

### miR-1-5p & TGF-βR1 are inversely expressed in the RV of the MCT treated rat

4 weeks after MCT or PBS-treatment, RV weight (RV / ((left ventricle) LV + septum)) and right ventricular systolic pressure (RVSP) were found to be higher in the MCT-treated rats compared to controls confirming increased vascular resistance, as previously published [18]. miR-1-5p expression was decreased 7-fold (p = 0.0077) in the RV of MCT compared to PBS treated rats (Fig 1A). There was no significant difference in expression of miR-1-5p in the left ventricles of the MCT and PBS-treated animals. Median expression of miR-1-5p was higher in the LV than RV of MCT treated animals but this difference did not reach statistical significance (p = 0.089). TGF-βR1 mRNA expression was higher in MCT-treated rat RVs compared to PBS controls (2.5-fold, p = 0.008), whereas no change in expression was noted between PBS-treated RVs to LVs (Fig 1B). TGF-βR1 protein expression was also higher (2.5-fold, p = 0.004) (Fig 1C) in the RVs of monocrotaline treated rats.

### miR-1-5p targets TGF-βR1

In silico analysis predicted one binding-site for miR-1-5p in the 3'UTR TGF-βR1 (ALK5) of both humans and rats (Fig 2A). The region of the 3' UTR of human ALK-5 containing this sequence was amplified by PCR and cloned into the 3'-UTR of EGFP in the vector pCAGG-S-EGFP to generate pCAGGS-EGFP-3T and the effect of miR-1-5p on EGFP expression was determined in LHCN-M2 cells. Transfection of miR-1-5p reduced EGFP expression from pCAGGS-EGFP-3T compared to control miR-mimic but did not affect the expression of EGFP from pCAGGS-EGFP (Fig 2B).

### miR-1 reduces TGF-βR1 mRNA & protein expression and signalling in vitro

To determine the effect of miR-1-5p on TGF-βR1 mRNA and protein, LHCN-M2 cells were transiently transfected with miR-1-5p mimic or control mimic. Transfection with the miR-1-5p mimic significantly reduced both TGF-βR1 mRNA and protein levels compared to transfection with the control mimic (Fig 2C and 2D). To determine whether TGF-β signalling was suppressed by miR-1-5p, a luciferase reporter with a TGF-β responsive promoter (CAGA)12 was used. Transfection with miR-1-5p reduced TGF-β stimulated reporter expression in response to both 1 and 5 ng/mL TGF-β, compared to control mimic transfection. However, there was no effect of the miRNA on basal reporter activity (Fig 2E).

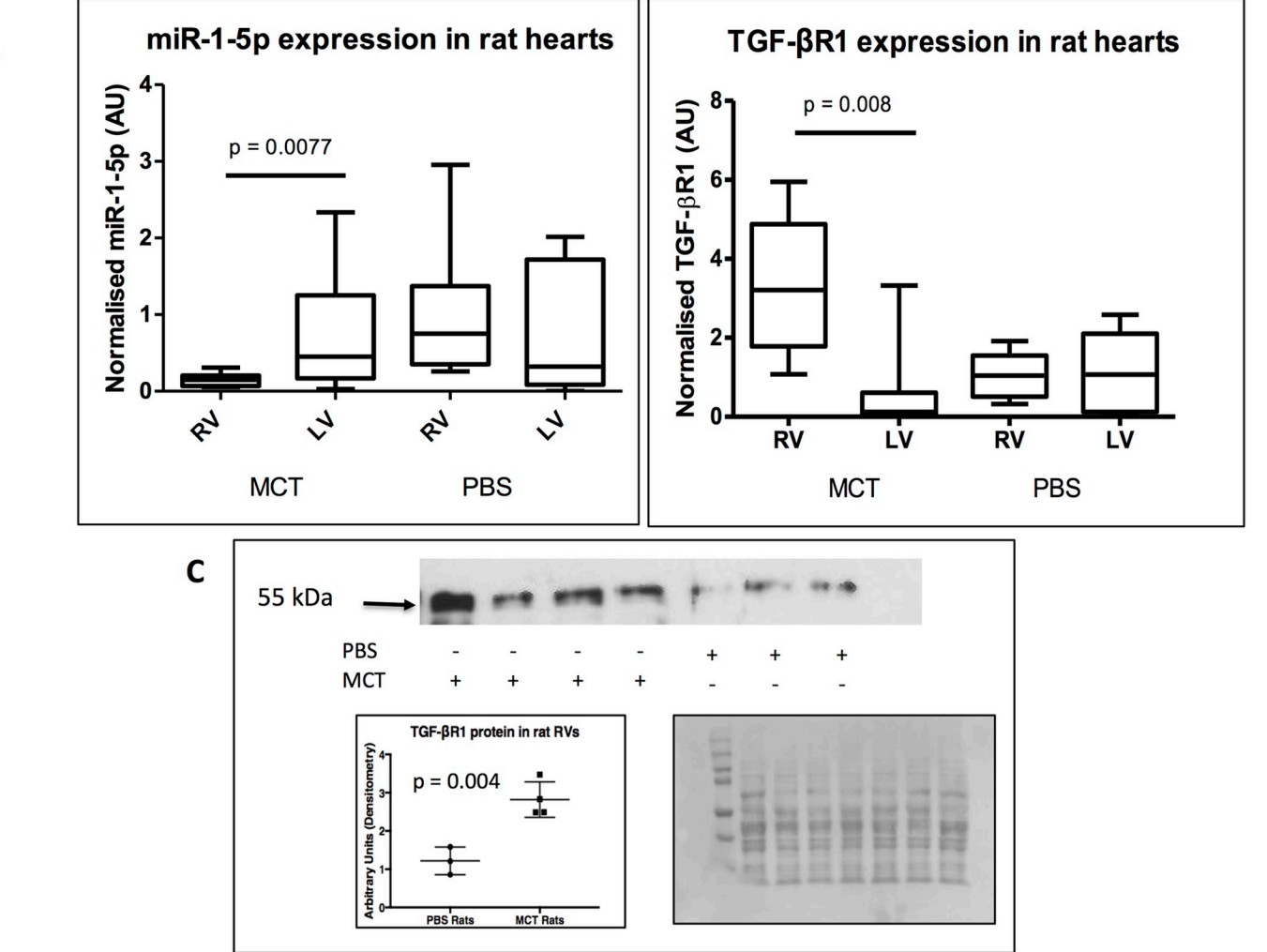

**Fig 1. miR-1-5p and TGF-βR1 are inversely expressed in the RV of MCT-treated rats with PAH.** miR-1 and transforming growth factor-beta receptor 1 (TGF-βR1) were quantified in RNA extracted from left (LV) and right ventricles (RVs) of monocrotaline (MCT)-treated rats by qPCR. A. miR-1-5p expression was significantly reduced in MCT-treated rat RVs compared to phosphate buffered saline (PBS)-treated (9 animals in each group). Mean expression of miR-1-5p was lower in the MCT RV compared to the MCT LV but this difference was not statistically significant. B. TGF-βR1 expression was significantly increased in MCT-treated rat RVs compared to LVs and to PBS-treated RVs. C. Western blot showing increased TGF-βR1 protein in the MCT-treated RVs compared to PBS treated RVs. Quantification of TGF-βR1 (bottom left) normalised to total protein in each lane as determined by Ponceau S staining (bottom right).

## Discussion

Our data identify miR-1 as a regulator of TGF-β signalling by suppressing the expression of ALK-5. Furthermore, the data show that miR-1-5p is suppressed in the RV of MCT rats coincident with increased ALK-5. This observation suggests that under normal conditions miR-1 reduces the sensitivity of cardiomyocytes to TGF-β by reducing the expression of ALK-5. Consequently, this reduction in miR-1-5p in hypertrophy leads to an increase in ALK-5 protein levels and thereby an increase in TGF-β signalling.

Given the effects of TGF-β on cardiac myocytes such an increase in TGF-β signalling is likely to contribute to cardiac hypertrophy. However, there are a number of limitations to our study that preclude us from confirming that the observed reduction in miR-1 promotes cardiac hypertrophy by relieving miR-1 suppression of TGF-β signalling. Firstly, we cannot determine whether the suppression in miR-1-5p preceded or came in the early phase of the

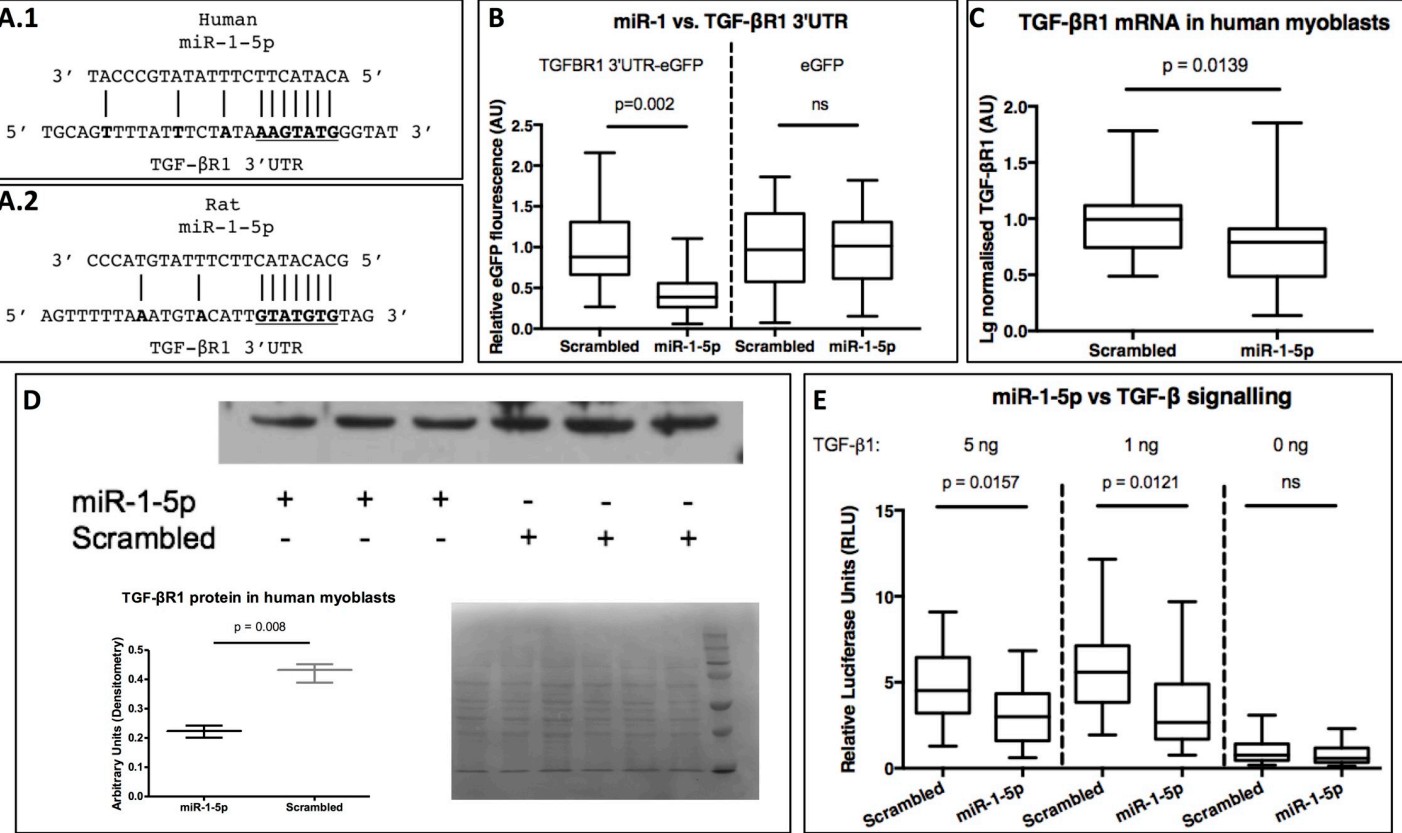

**Fig 2. miR-1 targets TGF-βR1 and inhibits TGF-β signalling.** A. Putative binding side of miR-1- 5p in the transforming growth factor-beta receptor 1 (TGF-βR1) 3'-untranslated region (3'UTR) in humans (A.1) and rats (A.2). Transfection of LHCN-M2 cells with miR1-5p significantly reduced enhanced green fluorescent protein (EGFP) expression from a reporter gene containing the TGF-βR1 3'UTR binding site (B). TGF-βR1 mRNA (C) and TGF-βR1 protein (D) are reduced following miR-1 transfection. E. Transfection of LHCN-M2 cells with miR-1-5p reduced TGF-β1 stimulated luciferase reporter gene expression.

hypertrophic response, which would be required for the reduction in miR-1 to be a key regulator of the response rather than an epiphenomenon. Secondly, we cannot be sure that the increase in ALK-5 protein occurred in the cardiac myocytes rather than in fibroblasts where it would contribute to fibrosis. For example, in pressure overload hypertrophy neutralizing antibodies to TGF-β 1 inhibit the fibrotic response but do not inhibit the hypertrophy of the myocytes [24]. This lack of effect may be due to a limited effect of TGF-β 1 in cardiac myocyte hypertrophy or may be due to the inability of the antibodies to interact with the pool of TGF-β 1 that acts on myocytes. Similarly, it may be a consequence of alternative TGF-β ligands (e.g. TGF-β 3) which also signal via TGF- βR1 contributing to the hypertrophic response and are not neutralized by the antibody, but which would be inhibited by over-expression of a dominant inhibitory TGF-β type II receptor (TGFBR2). Finally, miR-1 has been shown to affect the expression of a number of different proteins that regulate cardiac cell physiology in a manner consistent with the suppression of growth. Consequently, we cannot determine the relative importance of the interaction between miR-1 and TGF-βR1 in the hypertrophic response. These proteins include the sodium calcium exchanger (NCX1) [25] cyclin-dependent kinase 9 (Cdk9) and fibronectin [14] in the heart. Further evidence for a role for miR-1 in the development of the RV hypertrophy in response to PAH would require the delivery of miR-1 to the RV (e.g. by adenovirus) and demonstration that this slowed or prevented the development of hypertrophy. However, these experiments are beyond the current scope of our study and

whilst they have not been performed in PAH similar experiments have shown that over-expression of miR-1-5p reverses insulin-like growth factor-1 (IGF1) induced hypertrophy *in vitro* [26,27] and that adenoviral delivery of miR-1 to the myocardium has been shown to reverse overload induced LV hypertrophy [28]. Furthermore, other studies have shown increasing SERCA2a reverses hypertrophy and restores miR-1 expression [25]. Similarly, determination of the relative contribution of TGF-β signalling in the development of the same hypertrophy could be determined by the delivery of TGF-β neutralizing antibodies or more selectively by over-expression of a dominant inhibitory form of the TGFBR2 in the RV in a model of PAH.

miR-1-5p expression is not restricted to cardiomyocytes as it is also highly expressed in vascular and skeletal muscle cells. Consequently, it seems likely that miR-1 may also contribute to the TGF-β responses of these cells and thereby to TGF-β driven pathology in the associated tissues. Increased TGF-β signalling is suggested to promote the pulmonary artery smooth muscle cell (PASMC) proliferation that contributes to PAH [29]. Indeed, ALK-5 is thought to mediate abnormal PASMC proliferation in patients with familial PAH and inhibition of ALK-5 has been shown to inhibit the progression of PAH [30]. Similarly over-expression of a dominant inhibitory TGFBR2 has been shown to inhibit the development of PAH [31]. Consequently, it is possible that reduced expression of miR-1 in PASMCs contributes to this phenotype but appropriate changes in miR-1 expression in the PASMCs have yet to be demonstrated so such a contribution remains to be determined. In skeletal muscle, signalling through ALK-5 is a major contributor to skeletal muscle atrophy with both TGF-β promoting atrophy via ALK-5 and the TGF-βR2 and activin and myostatin promoting atrophy through ALK-5 and the activin type 2B receptor. Consistent with a role for the expression of miR-1-5p contributing to the maintenance of muscle mass, reduced miR-1 expression has been observed in the quadriceps of patients with intensive care unit-acquired weakness (ICUAW) and those with chronic obstructive pulmonary disease (COPD) [15,19]. In both conditions there is muscle atrophy and increased signalling through ALK-5.

In conclusion we have shown that miR-1-5p binds to the 3'UTR of the TGF-β1 receptor ALK-5, suppressing the expression of the protein and the strength of signals induced by the ligand. These data suggest that under basal conditions high miR-1 expression reduces TGF-βR1 and lowers the response of the cell to TGF-β ligands. However, in the RV of rats with PAH the reduction in miR-1 contributes to hypertrophy by increasing ALK-5 protein levels.

## Supporting information

**S1 Fig. The full western blot from Fig 1C.**
(TIFF)

**S2 Fig. Full blot for Fig 2D.**
(TIFF)

## Acknowledgments

We are grateful to Dr Vincent Mouly (UPMC, University of Paris) for the gift of the LHCN-M2 cell line.

## Author Contributions

**Conceptualization:** Martin Connolly, Stephen J. Wort, Paul R. Kemp.

**Data curation:** Martin Connolly, Benjamin E. Garfield.

**Formal analysis:** Martin Connolly, Paul R. Kemp.

**Funding acquisition:** Stephen J. Wort.

**Investigation:** Martin Connolly, Alexi Crosby.

**Methodology:** Martin Connolly, Benjamin E. Garfield, Alexi Crosby.

**Project administration:** Nick W. Morrell.

**Supervision:** Nick W. Morrell, Paul R. Kemp.

**Writing – original draft:** Martin Connolly, Paul R. Kemp.

**Writing – review & editing:** Alexi Crosby, Nick W. Morrell, Stephen J. Wort, Paul R. Kemp.

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
