## [Decision Letter · Decision Letter 0]

20 Nov 2019

PONE-D-19-27153

miR-1-5p targets TGF-βR1 and is suppressed in the hypertrophying hearts of rats with pulmonary arterial hypertension

PLOS ONE

Dear Dr. Kemp,

Thank you for submitting your manuscript to PLOS ONE. After careful consideration, we feel that it has merit but does not fully meet PLOS ONE’s publication criteria as it currently stands. Therefore, we invite you to submit a revised version of the manuscript that addresses the points raised during the review process.

Specifically, referee #1 correctly pointed out that to demonstrate miR-1 involvement in the development of cardiac hypertrophy with pulmonary arterial hypertension, miR-1 gain or loss of function experiments in vivo are indicated. These experiments are strongly recommended. Should the authors being unable to perform them, an accurate and detailed discussion of the consequent study limitations is required.

The authors are also requested to address all the other issues raised by both referees.

We would appreciate receiving your revised manuscript by 19/02/2020. To enhance the reproducibility of your results, we recommend that if applicable you deposit your laboratory protocols in protocols.io, where a protocol can be assigned its own identifier (DOI) such that it can be cited independently in the future. For instructions see: http://journals.plos.org/plosone/s/submission-guidelines#loc-laboratory-protocols

We look forward to receiving your revised manuscript.

Kind regards,

Fabio Martelli

Academic Editor

PLOS ONE

Journal Requirements:

2. Please provide additional information about the LHCN-M2 cell line used in this work, including source, history, culture conditions and any quality control testing procedures (authentication, characterisation, and mycoplasma testing). For more information, please see http://journals.plos.org/plosone/s/submission-guidelines#loc-cell-lines.

3. At this time, we ask that you please provide additional details regarding the monocrotaline drug used in this study. Specifically, please provide the source, product number and lot number of the drug. Thank you for your attention this request.

4. To comply with PLOS ONE submission guidelines, in your Methods section, please provide additional information regarding your statistical analyses. For more information on PLOS ONE's expectations for statistical reporting, please see https://journals.plos.org/plosone/s/submission-guidelines.#loc-statistical-reporting.

5. Your ethics statement must appear in the Methods section of your manuscript. If your ethics statement is written in any section besides the Methods, please move it to the Methods section and delete it from any other section. Please also ensure that your ethics statement is included in your manuscript, as the ethics section of your online submission will not be published alongside your manuscript.

Reviewers' comments:

Reviewer's Responses to Questions

**Comments to the Author**

1. Is the manuscript technically sound, and do the data support the conclusions?

Reviewer #1: Partly

Reviewer #2: Yes

2. Has the statistical analysis been performed appropriately and rigorously? 

Reviewer #1: Yes

Reviewer #2: Yes

3. Have the authors made all data underlying the findings in their manuscript fully available?

Reviewer #1: Yes

Reviewer #2: Yes

4. Is the manuscript presented in an intelligible fashion and written in standard English?

Reviewer #1: Yes

Reviewer #2: Yes

5. Review Comments to the Author

Reviewer #1: The study is of putative interest as it shows that miR-1-5p binds and suppress the TGF-β1 receptor ALK-5 expression thereby reduce TGF-β signalling, which may regulate cardiac hypertrophy. In an animal model of pulmonary arterial hypertension which is associated with right ventricle hypertrophy, the authors showed that miR-1 was reduced while TGF-βR1 protein levels increased. Using in vitro cell culture experiments they demonstrated that miR-1 suppress ALK5 and TGF β signalling.

The manuscript is logical and well-structured and the experiments have been described in sufficient detail. Nevertheless, although potentially interesting, the information brought forward by this study is too preliminary and there are some issues that must be addressed before being published.

Major issues:

This is a rather elementary and indirect demonstration of miR-1 in the development of cardiac hypertrophy caused by pulmonary arterial hypertension. In order the truly demonstrate that this axis is directly involved in the development of cardiac hypertrophy, animal studies using inhibition or overexpression of miR-1 (using Antagomirs or miR mimics) are needed. Moreover, histologically measured cardiomyocyte hypertrophy should be also performed.

Minor points:

1. The method section does not describe the pCAGGS-GFP assay.

2. Figures are reversed (Fig 1 si fig 2 and vice versa)

3. The connection between TGFbR1 inactivation data and the luciferase reporter assay is not clearly explained. There should be better stated that (CAGA)12-luciferase reporter plasmid is exclusively activated by TGF-β-induced complex between Smad3 and Smad4.

4. Statistical data for ALK-5 protein expression quantification should be included. Aslo the graph in 2D should be presented as Box-plot like all the others.

Reviewer #2: In this paper, the authors reported that miR-1 expression is reduced in the hypertrophic right ventricle of monocrotaline-treated rats. The authors identified by bioinformatic analysis that TGF-βR1 is a potential target for miR-1, the expression of which being increased in rat right ventricles. Finally, the authors used miR-1-mimic to reduce EGFP expression from a reporter vector containing the ALK5 3’-UTR and also knocked down endogenous TGF-βR1. The data confirm miR-1 targets TGF-βR1 thereby reduce TGF-β signaling, which may regulate cardiac hypertrophy. The article is interesting and original, with a good experimental design and bioinformatics approach. However, there are some issues to be solved that could improve the manuscript.

General comments:

1. Abstract need a clear aim or a motivation for the study presented; also, all abbreviations in the abstract must be detailed or avoided.

2. The authors should provide the “Supporting Information files” according to their statement: “All relevant data are within the manuscript and its Supporting Information files.”

3. Overall in the manuscript, the authors should provide all the abbreviations used when they are used for the first time. Many abbreviations are not detailed (see TGFβ, SMAD7, SMURF1, BMP, HDAC4, ALK5 at pages 3-4). Of special significance are the missing definition for RV and LV (pages 4 and 7) and MCT (pages 6-7 and later).

4. Methods, page 5: “RNA extraction and qPCR”. The authors should provide either the commercial ID or the custom sequences for all the primers used here for the SYBR green-based PCR for miRNA and mRNA expression (maybe in a Supplemental file, also missed) since these sequences are absent in the given reference #19.

5. Methods section missed the description of the in silico analysis approach performed for miR-1-5p and details about TGFβR1 3’UTR cloning approach (both results being also described at page 7).

6. The Figures 1 and 2 are given in the reverse order; please correct this.

7. When talking about data in Fig. 2A at page 7, it should be better to provide both Fig. 2A.1 and Fig. 2A.2 panels (there is no Fig. 2A there).

8. A minor typographical correction is needed to separate the third subtitle from the next paragraph from Results section. Also, at page 9, a correction is necessary for the references 15, 18 in the first paragraph.

9. Figure 1 legend needs a correction for its title: “miR-1-5p and TGF-βR1 are inversely expressed in the RV of PAH” could probably become “miR-1-5p and TGF-βR1 are inversely expressed in the RV of MCT-treated rats with PAH”.

10. Both Figure legends must be self-explanatory when describing the panels, with all abbreviations detailed, independently of the text manuscript.

6. PLOS authors have the option to publish the peer review history of their article (what does this mean?). If published, this will include your full peer review and any attached files.

Reviewer #1: No

Reviewer #2: Yes: Loredan S. Niculescu

---

## [Author Response · Author response to Decision Letter 0]

2 Jan 2020

Response to Referees’ comments

Specifically, referee #1 correctly pointed out that to demonstrate miR-1 involvement in the development of cardiac hypertrophy with pulmonary arterial hypertension, miR-1 gain or loss of function experiments in vivo are indicated. These experiments are strongly recommended. Should the authors being unable to perform them, an accurate and detailed discussion of the consequent study limitations is required.

Reviewer #1: The study is of putative interest as it shows that miR-1-5p binds and suppress the TGF-β1 receptor ALK-5 expression thereby reduce TGF-β signalling, which may regulate cardiac hypertrophy. In an animal model of pulmonary arterial hypertension which is associated with right ventricle hypertrophy, the authors showed that miR-1 was reduced while TGF-βR1 protein levels increased. Using in vitro cell culture experiments they demonstrated that miR-1 suppress ALK5 and TGF β signalling.

The manuscript is logical and well-structured and the experiments have been described in sufficient detail. Nevertheless, although potentially interesting, the information brought forward by this study is too preliminary and there are some issues that must be addressed before being published.

Major issues:

This is a rather elementary and indirect demonstration of miR-1 in the development of cardiac hypertrophy caused by pulmonary arterial hypertension. In order the truly demonstrate that this axis is directly involved in the development of cardiac hypertrophy, animal studies using inhibition or overexpression of miR-1 (using Antagomirs or miR mimics) are needed. Moreover, histologically measured cardiomyocyte hypertrophy should be also performed.

We accept the point that the reviewer raises and that it would be ideal to use gain and loss of function studies to demonstrate the relative importance of miR-1 in pulmonary hypertrophy as a consequence of PAH. Unfortunately we do not have either the funding or the licences required to carry out this experiment within the timeframe. However, as the editor has allowed us to detail the study limitations we have included a section describing the limitations of our study and the need to carry out similar experiments in a PAH model.

This section reads:

Given the effects of TGF-β on cardiac myocytes such an increase in TGF-β signaling is likely to contribute to cardiac hypertrophy. However, there are a number of limitations to our study that preclude us from confirming that the observed reduction in miR-1 promotes cardiac hypertrophy by relieving miR-1 suppression of TGF- β signaling. Firstly, we cannot determine whether the suppression in miR-1-5p preceded or came in the early phase of the hypertrophic response, which would be required for the reduction in miR-1 to be a key regulator of the response rather than an epiphenomenon. Secondly, we cannot be sure that the increase in ALK-5 protein occurred in the cardiac myocytes rather than in fibroblasts where it would contribute to fibrosis. For example, in pressure overload hypertrophy neutralizing antibodies to TGF-β 1 inhibit the fibrotic response but do not inhibit the hypertrophy of the myocytes (24). This lack of effect may be due to a limited effect of TGFβ 1 in cardiac myocyte hypertrophy or may be due to the inability of the antibodies to interact with the pool of TGF-β 1 that acts on myocytes. Similarly it may be a consequence of alternative TGFβ ligands (e.g. TGFβ 3) which also signal via TGFBR1 contributing to the hypertrophic response and are not neutralized by the antibody, but which would be inhibited by over-expression of a dominant inhibitory TGFβ type II receptor (TGFBR2). Finally, miR-1 has been shown to affect the expression of a number of different proteins that regulate cardiac cell physiology in a manner consistent with the suppression of growth. Consequently, we cannot determine the relative importance of the interaction between miR-1 and TGFβR1 in the hypertrophic response. These proteins include the sodium calcium exchanger (NCX1)(25) cyclin-dependent kinase 9 (Cdk9) and fibronectin (14) in the heart. Further evidence for a role for miR-1 in the development of the RV hypertrophy in response to PAH would require the delivery of miR-1 to the RV (e.g. by adenovirus) and demonstration that this slowed or prevented the development of hypertrophy. However, these experiments are beyond the current scope of our study and whilst they have not been performed in PAH similar experiments have shown that over-expression of miR-1-5p reverses insulin-like growth factor-1 (IGF1) induced hypertrophy in vitro (26,27) and that adenoviral delivery of miR-1 to the myocardium has been shown to reverse overload induced LV hypertrophy (28). Furthermore, other studies have shown increasing SERCA2a reverses hypertrophy and restores miR-1 expression (25). Similarly determination of the relative contribution of TGF-β signaling in the development of the same hypertrophy could be determined by the delivery of TGF-β neutralizing antibodies or more selectively by over-expression of a dominant inhibitory form of the TGFBR2 in the RV in a model of PAH. 

Minor points:

1. The method section does not describe the pCAGGS-GFP assay.

We apologise for this omission which has been corrected and the added section reads

EGFP reporter cloning & assay - A section of the 3’UTR of TGF-�R1 containing the putative miR-1 binding site was amplified from human cDNA using primers: forward (5’–3’), GGAGATCTGGGTGTTTGATATTTCTTCAT reverse: (5’-3’) GGGGGATCCGGACATTTTCTGTACATATCTTA and ligated into pGEM-T Easy (Promega). After sequencing to ensure the correct DNA had been amplified the insert was removed by BglII and BamHI digestion and ligated into a pCAGGS-enhanced green fluorescent protein (EGFP) vector down-stream of the coding region. Final plasmid sequences were confirmed by sequencing.

To quantify the effect of miR-1 on protein expression cells were transfected with the miR-mimic or control as previously described (21). Twenty four hours later, the cells were transfected with this pCAGGS-EGFP-reporter either with or without the 3’UTR inset, cells were then lysed 24h later and EGFP expression was quantified by fluorescence with excitation at 480nm and emission detected at 510nm in a Cytofluor plate reader (Applied Biosystems).

2. Figures are reversed (Fig 1 is fig 2 and vice versa)

The figures have now been placed in the correct order.

3. The connection between TGFbR1 inactivation data and the luciferase reporter assay is not clearly explained. There should be better stated that (CAGA)12-luciferase reporter plasmid is exclusively activated by TGF-β-induced complex between Smad3 and Smad4.

24 hours following miRNA transfection, cells were transfected with a 3:1 ratio of luciferase reporter vectors (CAGA)12 (firefly luciferase) and pRLTK (renilla luciferase) as previously described; the (CAGA)12 vector contains a SMAD binding element (SBE) specific to SMAD3 and SMAD4 (23).

4. Statistical data for ALK-5 protein expression quantification should be included. Also the graph in 2D should be presented as Box-plot like all the others.

The graph needs to be presented as it stands because there are only 3 data points in the Western blots. Consequently we can only show min/max and the median bar. This is the standard presentation of box plots with this number of data points in GraphPad.

Reviewer #2: In this paper, the authors reported that miR-1 expression is reduced in the hypertrophic right ventricle of monocrotaline-treated rats. The authors identified by bioinformatic analysis that TGF-βR1 is a potential target for miR-1, the expression of which being increased in rat right ventricles. Finally, the authors used miR-1-mimic to reduce EGFP expression from a reporter vector containing the ALK5 3’-UTR and also knocked down endogenous TGF-βR1. The data confirm miR-1 targets TGF-βR1 thereby reduce TGF-β signaling, which may regulate cardiac hypertrophy. The article is interesting and original, with a good experimental design and bioinformatics approach. However, there are some issues to be solved that could improve the manuscript.

General comments:

1. Abstract need a clear aim or a motivation for the study presented; also, all abbreviations in the abstract must be detailed or avoided.

We have reworded to the abstract to highlight our starting hypothesis and removed the bulk of the abbreviation. Those we couldn’t delete have been detailed. The abstract now reads as follows.

The microRNA miR-1 is an important regulator of muscle phenotype including cardiac muscle. Down-regulation of miR-1 has been shown to occur in left ventricular hypertrophy but its contribution to right ventricular hypertrophy in pulmonary arterial hypertension are not known. Previous studies have suggested that miR-1 may suppress transforming growth factor-beta (TGF-β) signalling, an important pro-hypertrophic pathway but only indirect mechanisms of regulation have been identified. We identified the TGF-β type 1 receptor (TGF-βR1) as a putative miR-1 target. We therefore hypothesized that miR-1 and TGF-βR1 expression would be inversely correlated in hypertrophying right ventricle of rats with pulmonary arterial hypertension and that miR-1 would inhibit TGF-β signalling by targeting TGF-βR1 expression. Quantification of miR-1 and TGF-βR1 in the right ventricle of rats treated with monocrotaline to induce pulmonary arterial hypertension showed that these was reduced in the hypertrophying right ventricle. A miR-1-mimic reduced enhanced green fluorescent protein expression from a reporter vector containing the TGF-βR1 3’- untranslated region and knocked down endogenous TGF-βR1. Lastly, miR-1 reduced TGF-β activation of a (mothers against decapentaplegic homolog) SMAD2/3-dependent reporter. Taken together, these data suggest that miR-1 targets TGF-βR1 and reduces TGF-β signaling, so a reduction in miR-1 expression may increase TGF-β signaling and contribute to cardiac hypertrophy.

2. The authors should provide the “Supporting Information files” according to their statement: “All relevant data are within the manuscript and its Supporting Information files.”

The initial file did not have any supplementary information and the statement was copied from the submission website. However, we have now included the full western blots as supplementary information as required by the journal.

3. Overall in the manuscript, the authors should provide all the abbreviations used when they are used for the first time. Many abbreviations are not detailed (see TGFβ, SMAD7, SMURF1, BMP, HDAC4, ALK5 at pages 3-4). Of special significance are the missing definition for RV and LV (pages 4 and 7) and MCT (pages 6-7 and later).

We have added these abbreviations in the relevant places

4. Methods, page 5: “RNA extraction and qPCR”. The authors should provide either the commercial ID or the custom sequences for all the primers used here for the SYBR green-based PCR for miRNA and mRNA expression (maybe in a Supplemental file, also missed) since these sequences are absent in the given reference #19.

We apologise for this omission and have added a table with the primer sequences used. 

The reverse primer for the PCR of miR-1 and U6 was the reverse primer provided with the polyadenylation and first strand synthesis kit.

Table 1: Primer list for qPCR

Gene Forward sequence Reverse sequence

miR-1-5p CCGGTGGAATGTAAAGAAGTATGTAT Agilent Universal reverse primer

U6 (housekeeper) CTCGCTTCGGCAGCACA Agilent Universal reverse primer

TGF-βR1 GAACTCCCAACTACAGAAAAGCA GCAGACTGGACCAGCAATGA

GAPDH (housekeeper) GGTGGTCTCCTCTGACTTCAACA GTTGCTGTAGCCAAATTCGTTGT

5. Methods section missed the description of the in silico analysis approach performed for miR-1-5p and details about TGFβR1 3’UTR cloning approach (both results being also described at page 7).

We have added the following to describe the in silico analysis :

In silico analysis of miR-1 – Potential targets in the TGF-β signaling pathway for miR-1 were identified via the miRWalk 2.0 database, (17). Putative targets were identified but screening the data base for targets identified by all 10 of the available algorithms. Putative targets were chosen as those identified by 5 or more algorithms. TGF-βR1/ALK-5 was predicted as a target by 6 algorithms including miTRWalk, miRanda and Targetscan. As TGF-β signaling has previously been shown to be important in regulating cardiac size, it was chosen as the basis of an enquiry into miR-1 regulation of cardiac hypertrophy.

We have also described our cloning of the 3’UTR into the EGFP plasmid as indicated in the response to Reviewer 1

6. The Figures 1 and 2 are given in the reverse order; please correct this.

We apologise for the oversight and have corrected this error

7. When talking about data in Fig. 2A at page 7, it should be better to provide both Fig. 2A.1 and Fig. 2A.2 panels (there is no Fig. 2A there).

We have made this addition

8. A minor typographical correction is needed to separate the third subtitle from the next paragraph from Results section. Also, at page 9, a correction is necessary for the references 15, 18 in the first paragraph.

Corrected

9. Figure 1 legend needs a correction for its title: “miR-1-5p and TGF-βR1 are inversely expressed in the RV of PAH” could probably become “miR-1-5p and TGF-βR1 are inversely expressed in the RV of MCT-treated rats with PAH”.

Corrected

10. Both Figure legends must be self-explanatory when describing the panels, with all abbreviations detailed, independently of the text manuscript.

 Corrected

Response to Editorial Queries 

We have corrected these as requested

2. Please provide additional information about the LHCN-M2 cell line used in this work, including source, history, culture conditions and any quality control testing procedures (authentication, characterisation, and mycoplasma testing). For more information, please see http://journals.plos.org/plosone/s/submission-guidelines#loc-cell-lines.

We have added a section to cover this point as follows

LHCN-M2 human skeletal myoblasts obtained from Vincent Mouly (Sorbonne Unitversité) in 2014 and were cultured in Skeletal muscle growth medium (PromoCell) supplemented with 20% FCS

3. At this time, we ask that you please provide additional details regarding the monocrotaline drug used in this study. Specifically, please provide the source, product number and lot number of the drug. Thank you for your attention this request.

We have added the supplier and product number. (MCT, Sigma; C2401) Unfortunately we do not have the lot number of this compound so cannot add this.

4. To comply with PLOS ONE submission guidelines, in your Methods section, please provide additional information regarding your statistical analyses. For more information on PLOS ONE's expectations for statistical reporting, please see https://journals.plos.org/plosone/s/submission-guidelines.#loc-statistical-reporting.

We have amended the statistics section as follows:

All statistical analyses were performed in GraphPad PRISM and no samples were excluded as outliers and data were analysed using a between samples design. Animal experiments were conducted in two groups for a total of 9 MCT- and 9 PBS-treated animals. Differences in miRNA and mRNA expression between animal groups and left and right ventricles were calculated via Kruskal-Wallis test (ANOVA) for non-parametric data with post-hoc analysis using Dunn’s multiple test correction. All other comparisons calculated using Student’s t-test for normally distributed data or by Mann-Whitney U test for non-parametric data. In vitro mRNA expression and luciferase data shown were produced in three independent experiments, each consisting of six independent transfections; mRNA measures assayed in duplicate. Box-plots expressed as median with min-max bars. In vitro protein expression data shown via western blot are three independent experiments from 6-well plates. All tests were based on two tailed analysis and differences were taken to be significant at p<0.05.

5. Your ethics statement must appear in the Methods section of your manuscript. If your ethics statement is written in any section besides the Methods, please move it to the Methods section and delete it from any other section. Please also ensure that your ethics statement is included in your manuscript, as the ethics section of your online submission will not be published alongside your manuscript.

The ethics statement is in the Methods section

We have added the uncropped and unadjusted images of the blots as Supporting images files and added a comment to the Methods section that this is where they are.

---

## [Editor Report · Decision Letter 1]

6 Feb 2020

miR-1-5p targets TGF-βR1 and is suppressed in the hypertrophying hearts of rats with pulmonary arterial hypertension

PONE-D-19-27153R1

Dear Dr. Kemp,

We are pleased to inform you that your manuscript has been judged scientifically suitable for publication and will be formally accepted for publication once it complies with all outstanding technical requirements.

With kind regards,

Fabio Martelli

Academic Editor

PLOS ONE
---

## [Editor Report · Acceptance letter]

14 Feb 2020

PONE-D-19-27153R1 

miR-1-5p targets TGF-βR1 and is suppressed in the hypertrophying hearts of rats with pulmonary arterial hypertension 

Dear Dr. Kemp:

I am pleased to inform you that your manuscript has been deemed suitable for publication in PLOS ONE. Congratulations! Your manuscript is now with our production department. 

With kind regards,

on behalf of

Dr. Fabio Martelli 

Academic Editor

PLOS ONE